# A Switching Hybrid Dynamical System: Toward Understanding Complex Interpersonal Behavior

**Yuji Yamamoto** [1,*,†]**, Akifumi Kijima** [2,†]**, Motoki Okumura** [3,†]**, Keiko Yokoyama** [1,†] **and Kazutoshi Gohara** [4,†]

1   Research Center of Health, Physical Fitness, and Sports, Nagoya University, Nagoya 464-8601, Japan; yokoyama@htc.nagoya-u.ac.jp
2   Graduate School of Education, University of Yamanashi, Kofu 400-8510, Japan; akijima@yamanashi.ac.jp
3   Graduate School of Education, Tokyo Gakugei University, Tokyo 184-8501, Japan; okumura@u-gakugei.ac.jp
4   Nano Biotechnology, Research Faculty of Engineering, Hokkaido University, Sapporo 060-0808, Japan; gohara@eng.hokudai.ac.jp
*   Correspondence: yamamoto@htc.nagoya-u.ac.jp; Tel.: +81-52-789-3964
†   These authors contributed equally to this work.

**Abstract:** Complex human behavior, including interlimb and interpersonal coordination, has been studied from a dynamical system perspective. We review the applications of a dynamical system approach to a sporting activity, which includes continuous, discrete, and switching dynamics. Continuous dynamics identified switching between in- and anti-phase synchronization, controlled by an interpersonal distance of 0.1 m during expert kendo matches, using a relative phase analysis. In the discrete dynamical system, return map analysis was applied to the time series of movements during kendo matches. Offensive and defensive maneuvers were classified as six coordination patterns, that is, attractors and repellers. Furthermore, these attractors and repellers exhibited two discrete states. Then, state transition probabilities were calculated based on the two states, which clarified the coordination patterns and switching behavior. We introduced switching dynamics with temporal inputs to clarify the simple rules underlying the complex behavior corresponding to switching inputs in a striking action as a non-autonomous system. As a result, we determined that the time evolution of the striking action was characterized as fractal-like movement patterns generated by a simple Cantor set rule with rotation. Finally, we propose a switching hybrid dynamics to understand both court-net sports, as strongly coupled interpersonal competition, and weakly coupled sports, such as martial arts.

**Keywords:** interpersonal coordination; competition; dynamical systems; discrete dynamics; continuous dynamics; sporting activity

---

## 1. Introduction

Exploring complex human behavior from the perspective of a dynamical system began with a historic experiment [1]. Three decades later, a new approach was developed using dynamical system theory and utilities to analyze and simulate complex phenomena, such as human movement, using super-computers. In this article, we review the applications of a dynamical system approach to human behavior, with particular regard to sports activities that require quick decision-making and appropriate execution. We provide applications for continuous dynamics, discrete dynamics, switching dynamics, and switching hybrid dynamics, with a short theoretical review.

Human behavior, particularly the behavior of rhythmic cyclical movements, can be understood as continuous dynamics that can be described using differential equations. In 1665, the Dutch physicist

Christiaan Huygens synchronized two pendulum clocks using the interactions of tiny vibrations in their common support [2,3]. Synchronizing the two clocks was regarded as entrainment and/or synchronization of the two coupled nonlinear oscillators model. The theoretical Haken-Kelso-Bunz (HKB) model describes the phase transition during the coordinating pattern observed between hands based on synergetics [4] and the nonlinear oscillator theory [5]. This model has been applied to intrapersonal and interpersonal coordination [6,7].

Furthermore, group synchrony or coordination has been examined using the Kuramoto model and/or the Kuramoto order parameter [8], with regard to synchronized clapping [9] and phase synchronization among rocking chairs in a small group [10,11]. Anti-phase synchronization of the tree frogs was analyzed as a phase oscillator model [12]. As another approach, the symmetric Hopf bifurcation theory based on group theory [13] was applied to investigate the synchronized pattern of three people during a sports activity [14]. These approaches could be regarded as oscillator dynamics, or continuous dynamics.

On the other hand, discrete dynamics and/or mapping can be depicted as iterated maps, that is, in terms of difference equations describing the time evolution, such as population dynamics with generations. The logistic map is a well-known map of how complex, chaotic behavior can arise from very simple quadratic difference equations. A Markov chain in discrete time, which is characterized as a state transition based on the probability distribution of the next state, depends only on the current state as a kind of discrete dynamics, and has been applied to baseball batting [15] and to squash [16,17]. However, these studies examined predictions pertaining to competitive sports performance and did not explore the underlying dynamics. We introduce examples to explore the simple rules underlying complex human behavior using a Lorenz map, that is, a first return map and a Poincaré map to convert continuous dynamics to discrete dynamics.

Previous studies have been based on the two coupled nonlinear oscillators model. Thus, these systems are closed systems with no abrupt changes in external input. We introduce the application of switching dynamics with temporal input to continuous striking movements. This model also uses the idea of the Poincaré map. Finally, to explore complex human movement as a dynamical system, we introduce switching hybrid dynamical systems, which include discrete and continuous dynamics with a feedback loop.

## 2. Continuous Dynamics

### 2.1. The Haken-Kelso-Bunz (HKB) Model

Human behavior, particularly rhythmic cyclical movement, is a phenomenon of entrainment and/or synchronization of two-coupled nonlinear oscillators [1]. In a well-known experiment, participants were asked to oscillate their index fingers at a common frequency. The relative phase, namely the difference between the oscillation phases of two fingers, depicts the spatiotemporal pattern of coordination as an order parameter. The coordination mode shows only two steady states: 0° relative phase (in-phase mode) and 180° relative phase (anti-phase mode). As movement frequency increases, the anti-phase spontaneously switched to the in-phase mode. However, the latter did not switch to the anti-phase mode. Below a critical frequency, the system showed bistable behavior, with both in- and anti-phase modes. However, above a critical frequency, the system showed monostable behavior with only the in-phase mode. The HKB model was developed based on synergetics [4] to explain the phase transition, including hysteresis, critical slowing down, and critical fluctuations using order and control parameters [5,18]. The equation revealed the potential function underlying the spontaneous transition from bistable to monostable behavior. Furthermore, the HKB model for intrapersonal coordination was extended to interpersonal coordination [6,7].

In the interpersonal coordination experiment, cross-spectral coherence and the distribution of the relative phase region were calculated as coordination indices [19–22]. Cross-spectral coherence provides a method to correlate two time series over a range of possible component frequencies. In other

words, this analysis reveals the dominant frequencies and/or the strength of the coordination. On the other hand, evaluating the relative phase distribution indicates which relative phase locations are attractive by identifying the dominant kind of coordination. In this analysis, the distribution of relative phase angles across nine relative phase regions between 0° and 180° was determined by calculating the frequency of occurrence of the relative phase angles in each of these regions.

Both coordination indices postulate periodic movement, such as that of a hand-held pendulum [21,22]. However, the behavior of competitive sports players does not always show such periodic movement. A cross-spectral analysis could not be applied to this kind of time series because of the abrupt changes and aperiodic behavior. In addition, the relative phase was usually calculated by point estimates [23] or continuous estimates [24]. However, an aperiodic or arbitrary time series cannot be calculated using these estimates. To solve this problem, the Hilbert transform [25,26] was applied to the aperiodic coordination signal in competitive sports to calculate the instantaneous phase.

Then, a relative phase analysis using the Hilbert transform was applied to court-net sports, such as tennis [27–29] and squash [30]. They reported that players' movements frequently switched between in- and anti-phase synchronization in the direction of the short axis of the court as they took turns hitting the ball. In addition, the speed scalar has been used as a collective variable to describe different patterns during badminton [31]. In these court-net sports, the ball is considered to be a physical link constraining the opponent's movement. These systems of two and/or more players can be regarded as strongly coupled oscillators [32]. However, it remains unclear whether each player and/or oscillator could be considered a self-excited oscillation or forced oscillation in court-net sports.

## 2.2. Identity of Synchronization Modes Revealed by the Relative Phase Region During Interpersonal Competition

As mentioned above, ball movement during court-net sports constrains player movement. However, in martial arts, such as boxing or fencing, the two players are not joined together physically and can move around freely, although each must co-adapt to the opponent's movement. Consequently, the movements of the players can be regarded as self-excited oscillations, and a system of two players can be considered a weakly coupled nonlinear oscillator [8,26,33]. We applied the distribution from a relative phase analysis to Japanese fencing, which is called kendo, to examine the synchronization mode during matches [34].

We observed 12 kendo matches among six expert members of a university kendo team. All matches followed official kendo rules. The candidate order parameter was the relative phase angle of the step toward-away velocity, and the candidate control parameter was the interpersonal distance between the two players. The interpersonal distance, called *ma'ai*, is crucial in kendo, because a player must simultaneously strike an opponent and avoid the opponent's counterstrike. Scoring a point (*ippon*) requires an accurate strike on an opponent. Consequently, the players must repeat step toward-away movements to adjust *ma'ai*, and a striking movement, which requires less than 0.4 s [34]. In this experiment, when both players stepped toward or away simultaneously, the coordination pattern was defined as in-phase coordination. In contrast, when one player stepped toward and another stepped away, it was defined as anti-phase coordination.

The results showed that anti-phase synchronization was clearly dominant at an interpersonal distance of less than 2.7 m (near distance). If player A moved toward player B, player B moved away from player A. However, in-phase synchronization was dominant when interpersonal distance exceeded 3.0 m (far distance). We focused on the 2.7–3.0 m distance to analyze the phase transition. The frequency of the relative phase in the 0.1 m range is shown in Figure 1. The results clearly show that higher anti-phase synchronization occurred at 2.7–2.8 m than at other distances, and this switched to in-phase synchronization at 2.9–3.0 m, setting a boundary at 2.8–2.9 m. This switching range (2.8–2.9 m) was slightly longer than the averaged possible striking distance, in line with the not-to-lose strategy [34]. This finding indicates that the players perceived and understood the need for minute changes on a 0.1 m scale and, consequently, regularly switched their movement to the appropriate direction or synchronization mode.

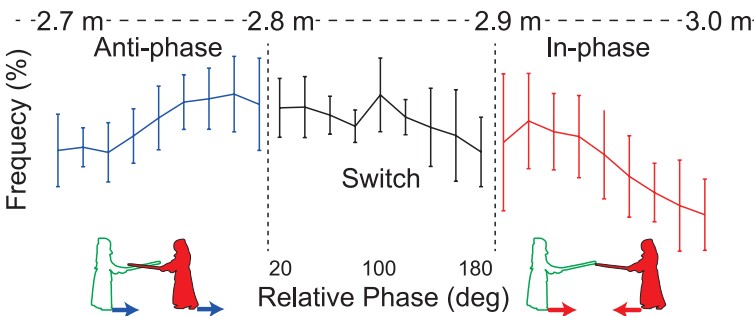

**Figure 1.** Abrupt changes in the interpersonal coordination pattern corresponding to an interpersonal distance of 2.7–3.0 m while playing kendo. The frequencies of the relative phase per 0.1 m interval at interpersonal distances of 2.7–3.0 m were calculated, and the means and standard deviations are presented. The relative phases were divided into nine ranges (0°–20°, 20°–40°, 40°–60°, 60°–80°, 80°–100°, 100°–120°, 120°–140°, 140°–160°, and 160°–180°). Anti-phase coordination was dominant at interpersonal distances of 2.7–2.8 m. However, in-phase coordination was dominant at interpersonal distances of 2.9–3.0 m. Modified from [34].

The distribution of the relative phase analysis revealed switching between the synchronization modes of interpersonal competition corresponding to the interpersonal distance as the control parameter. In addition, we examined the learning process in interpersonal competition via the play-tag game [35]. The relative phase analysis revealed that the synchronization modes changed from in-phase to anti-phase coordination by learning, because the players learned the not-to-lose strategy. However, in this analysis, the distribution of the relative phases across nine 20° relative phase regions from 0° to 180° was calculated. This means that the tendencies in the pooled data can be revealed, but the time evolution in the time series data cannot. In other words, while the coordination modes can be described, the coordination patterns in a shorter time window cannot. Interpersonal competition is characterized by aperiodic movements.

## 3. Continuous to Discrete Dynamics: Return Map

### 3.1. Lorenz Map

To describe the time evolution for interpersonal coordination, we introduce a method to explore regularity by reducing the dimensionality of continuous dynamics (e.g., three-dimensional flow) to discrete dynamics (e.g., two- or one-dimensional). The most popular method is the Lorenz map, which was a seminal work that contributed to the foundation of the chaos theory [36].

The meteorologist, Edward Lorenz, developed a simplified mathematical model for atmospheric convection [36]. The model is a system of three ordinary differential equations now known as the Lorenz equations:

$$\frac{dx}{dt} = \sigma(y - x) \tag{1}$$

$$\frac{dy}{dt} = x(r - z) - y \tag{2}$$

$$\frac{dz}{dt} = xy - bz \tag{3}$$

Here $x$, $y$, and $z$ are proportional to the intensity of convection motion, the temperature difference between the ascending and descending currents, and distortion of the vertical temperature profile from linearity, respectively. $\sigma$, $r$, and $b$ are the system parameters. Lorenz focused on a "single feature ([36], p. 138)", that is, $z_n$, of his three-dimensional strange attractor (Figure 2a). Figure 2b shows a time series of $z(t)$, and the $z(t)$ peaks were plotted as $z_n$ vs. $z_{n+1}$ (Figure 2c). The function $z_{n+1} = f(z_n)$ is called the Lorenz map. The Lorenz map shows the road to chaos through the bifurcation [33].

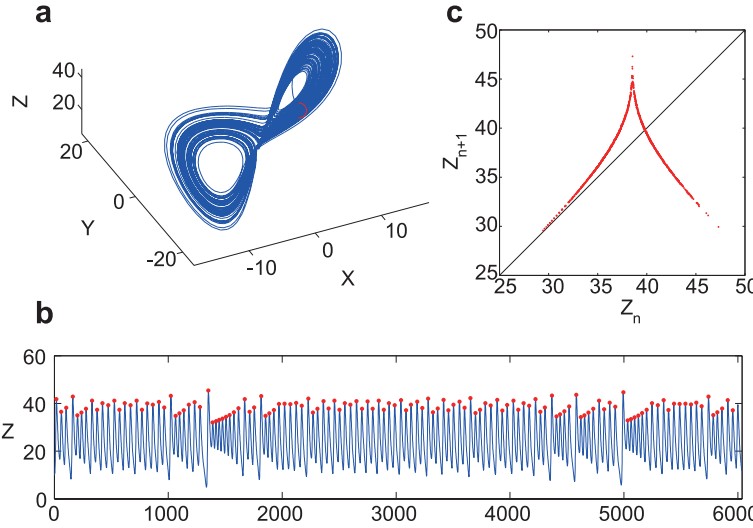

**Figure 2.** Lorenz attractor and Lorenz map. Parameters $\sigma = 10$, $b = 8/3$, $r = 28$, and $x_0 = y_0 = z_0 = 10$. (**a**) Lorenz attractor in three-dimensional space. (**b**) Time series of $z(t)$ and peaks of $z(t)$ are shown as red dots. (**c**) Lorenz map plotted as $z_n$ vs. $z_{n+1}$.

### 3.2. Identification of Coordination Patterns by the Return Map during Interpersonal Competition

The first return (Lorenz) map was applied to the offensive and defensive maneuvers that occur during kendo matches as interpersonal competition [37]. Figure 3a shows the time series of state variables comprising interpersonal distance and its velocity. We detected the peaks of the time series for interpersonal distance, and the peaks of the state variables were plotted as a return map as the present peak $X_n$ vs. the next peak $X_{n+1}$ (Figure 3b).

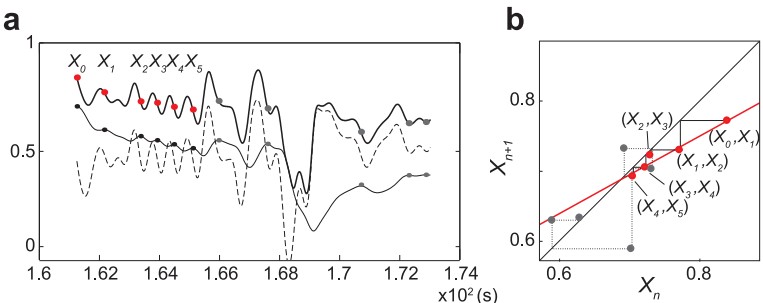

**Figure 3.** Procedure for depicting the return map from the time series of state variables. (**a**) Gray, broken, and black lines show the time series for normalized $X_{IPD}(t)$, normalized $V_{IPD}(t)$, and $X(t)$, respectively, for a 12-s trial with more than five peaks. The red and gray circles indicate the corresponding values of $X(t)$ for the peaks of $X_{IPD}(t)$. (**b**) Return map of the time series for the observed data, $X_n$ vs. $X_{n+1}$ using the amplitude of $X(t)$ at the peaks of $X_{IPD}(t)$ corresponding to the series of points (red and gray circles) in the panel shown in **a**. Modified from [37].

The periodicities on such a plot are the intersections with the identity line $X_n = X_{n+1}$. These intersections are known as attractive fixed points and repellers or saddle points. These attractive fixed points are deterministically approached from a direction called the stable direction or manifold, and the repellers diverge from these attractive fixed points along the unstable direction or manifold as a linear function. Theoretically, we postulated the linear function, $X_{n+1} = aX_n + b$. The intersections can be classified into two properties depending on the absolute value of $a$. When the absolute value of $a$ is less than 1, $|a|$ is less than 1. Then the intersection is considered to be an attractive fixed point (i.e., an "attractor"). When the absolute value of $a$ exceeds 1, $|a|$ exceeds 1. Then the intersection is referred to as a repellent fixed point (i.e., a "repeller"). An attractor can be further classified into two types.

When $0 < a < 1$, the trajectories are asymptotically close to the attractor (Figure 4a). When $-1 < a < 0$, the trajectories are rotationally close to the attractor (Figure 4b). A repeller also has two types of trajectories: $1 < a$, and $a < -1$, corresponding to asymptotical and rotational trajectories, respectively, as shown in Figure 4c,d. Trajectories approach and diverge from points that do not cross the line $X_n = X_{n+1}$. We postulated that this exponential function, $X_{n+1} = b\, exp(aX_n)$, and this logarithmic function, $X_{n+1} = a\, log X_n + b$ (Figure 4e), represent intermittency.

As shown in Figure 4, all six predicted types of functions were confirmed in the kendo matches, suggesting that the complex offensive and defensive maneuvers were generated by simple rules. Furthermore, we detected several functions or patterns during one offensive and defensive scene (Figure 4f), indicating that interpersonal coordination switched among several patterns during the competition.

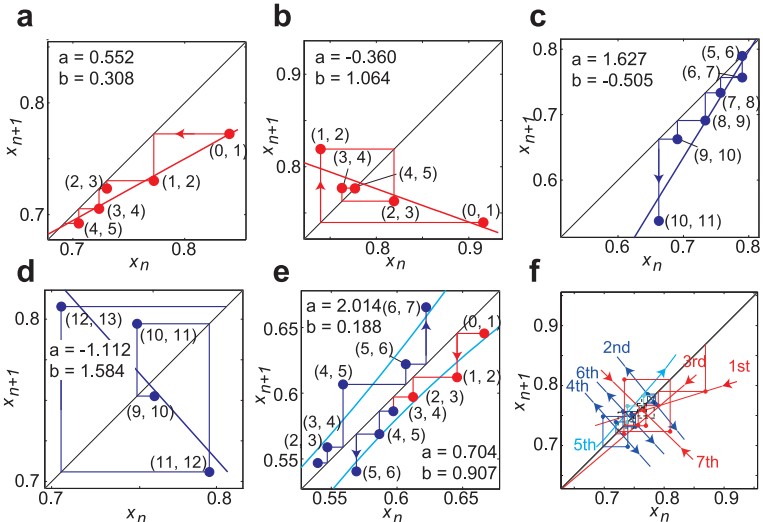

**Figure 4.** Examples of a well-fitted series of points by each function using the return map analysis. (**a–d**) Linear functions, $X_{n+1} = aX_n + b$, with four different slopes for $0 < a < 1, -1 < a < 0, 1 < a$, and $a < -1$, respectively. (**e**) Exponential function, $X_{n+1} = b\, exp(aX_n)$, and logarithmic function, $X_{n+1} = a\, log(X_n) + b$. (**f**) Examples of switching functions in one scene. The red lines show attractors, blue lines show repellers, and cyan lines show intermittencies. Modified from [37].

### 3.3. Identification of Switching Pattern for Expertise via the State Transition Probability

Figure 4 shows that the offensive and defensive maneuvers could be classified into six coordination patterns and that these patterns switched over short time scales. To clarify the characteristics of the expert and intermediate switching patterns, the histograms of the return maps for each group were calculated. As a result, we identified two discrete states in each histogram: the "farthest apart" high-velocity state (F) and the "nearest (closest) together" low-velocity state (N). We identified four trajectories, namely $\{X_n = F, X_{n+1} = F\}$, $\{X_n = N, X_{n+1} = N\}$, $\{X_n = F, X_{n+1} = N\}$, and $\{X_n = N, X_{n+1} = F\}$, as second-order transitions. Figure 5 shows the second-order state transition diagrams for experts and intermediates.

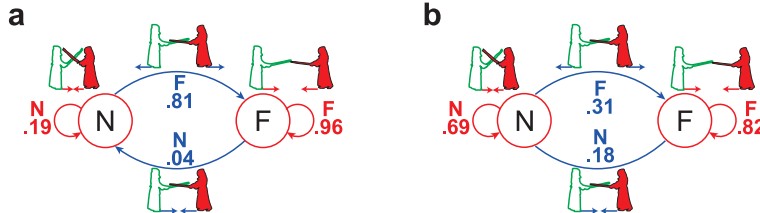

**Figure 5.** Second-order state transition diagrams with conditional probabilities consisting of the "farthest apart" high velocity states (F) and the "nearest together" low velocity state (N) for expert (**a**) and intermediate (**b**) competitors, respectively. Modified from [37].

The conditional probabilities for second-order state transitions of the experts were $\{Pr(F|F) = 0.96, Pr(N|F) = 0.04\}$, and $\{Pr(N|N) = 0.19, Pr(F|N) = 0.81\}$. The probabilities for the intermediates were $\{Pr(F|F) = 0.82, Pr(N|F) = 0.18\}$, and $\{Pr(N|N) = 0.69, Pr(F|N) = 0.31\}$. The transition probabilities between the experts and intermediates were clearly different; the experts were more often in the "farthest apart" high-velocity F-state. In contrast, the intermediate players generally remained in the "nearest (closest) together" low-velocity N-state.

The return map analysis revealed that the coordination patterns repeated over short time scales during interpersonal competitive behavior. In addition, the state transition probability analysis revealed differences in the switching patterns between experts and intermediates. Nevertheless, all patterns were shown for both levels of players. However, we considered two players as one system in these analyses. In other words, the two players were an autonomous, self-excited system. As a result, this approach clarified how the coordination pattern between the two players behaved rather than how the individuals behaved. The manner of the individual's adaptation to abrupt changes in the external environment must be examined for the individual perspective.

## 4. Continuous to Discrete Dynamics: Switching Dynamics

### 4.1. Switching Dynamics

The behavior of the individual is generated based on the behavior of other individuals during interpersonal coordination. Thus, a model with temporal external input must be considered to describe the behavior of the individual who adapts to changes in other individuals and/or the environment. In other words, the behavior of the individual must be considered a non-autonomous, excited system with external input. We applied switching dynamics to describe the behavior of the individual corresponding to temporal input [38,39].

A Poincaré map was used to simplify the analysis of the differential equation by reducing it to an iterated map [33]. A periodic trajectory with initial conditions within a section of the phase space leaves that section, and the point at which this trajectory first returns to that section is determined. This section is called the Poincaré section. Three-dimensional flow maps the two-dimensional Poincaré section, and the Poincaré map can be analyzed to understand the characteristics of the original system. A Poincaré map is a discrete dynamical system with a phase space that is one dimension smaller than the original continuous dynamical system. Switching dynamics applies the Poincaré map to reduce the dimensionality of continuous dynamics.

Figure 6a shows the trajectories for three periodic inputs: $I_1^*$, $I_2^*$, and $I_3^*$. These inputs are changed periodically to different amplitudes with the same period. Three limit cycle attractors, that is, excited attractors, $A_1$, $A_2$, and $A_3$, are observed in the cylindrical phase space $\mathcal{M}$. When the three inputs are switched stochastically in the system with finite intervals, the trajectories switch corresponding to the input (Figure 6b). It is known that these trajectories spread out around the excited attractors with a fractal-like structure. As a result, the Poincaré section shows the Sierpinski gasket (Figure 6c). All trajectories are considered to represent the transition between the excited attractors, called the fractal

transition between the excited attractors, to characterize the dynamics of the dissipative dynamical system excited by the temporal inputs. The details of this model have been explained in [38,39].

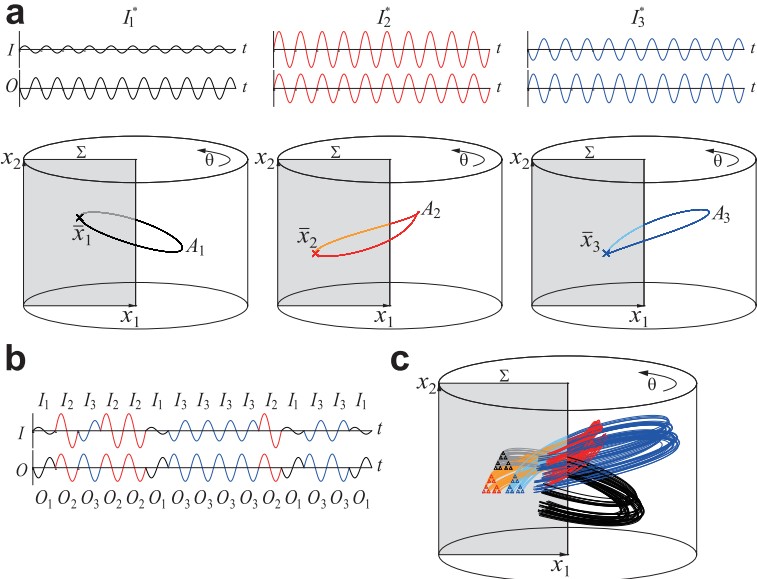

**Figure 6.** (**a**) Examples of time series for three periodic inputs. *I* and *O* denote the input and output time series, respectively. The trajectories for three periodic inputs in three-dimensional cylindrical phase space, $(x_1, x_2, \theta) \in \mathcal{M} : R^2 \times S^1$, corresponding to the colored trajectories denoted by $A_1$, $A_2$ and $A_3$ cross the Poincaré section $\Sigma : R^2$ at $\bar{x}_1$, $\bar{x}_2$ and $\bar{x}_3$, respectively. (**b**) An example of a time series for switching inputs. (**c**) The trajectories of randomly switching inputs and the cross points on the Poincaré section show the Sierpinski gasket as a result of the fractal transitions. Modified with permission from [39], Fractals 1999.

### 4.2. Underlying Simple Rule for Complex Striking Actions as per the Poincaré Map

We applied these switching dynamics to the striking action during tennis to understand complex human movements [40]. Two kinds of trajectories occurred when the ball was launched to the forehand or backhand side repeatedly as a periodic input condition. The trajectories were termed excited attractors, according to the input type (Figure 7a). When the ball was launched to the forehand and backhand sides randomly as a switching input condition, the trajectories in the cylindrical phase space were more complex (Figure 7b).

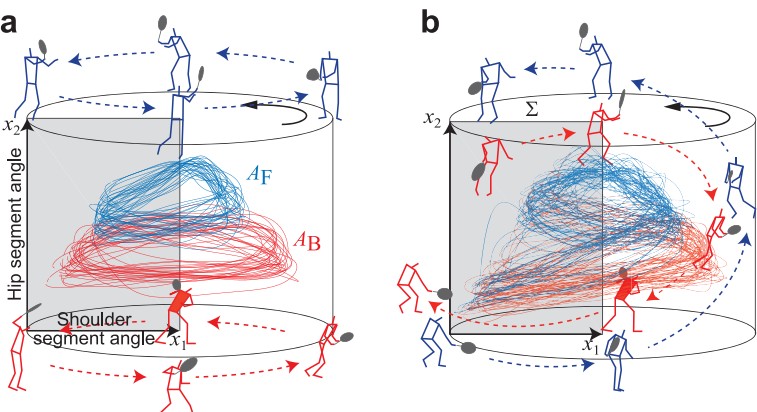

**Figure 7.** Trajectories in three-dimensional cylindrical phase space. (**a**) Periodic input condition, (**b**) switching input condition. The stick pictures show forehand and backhand striking movements at each point in the time series. Modified with permission from [40], Hum. Mov. Sci. 2000.

To understand the behavior of the system, we examined the Poincaré map on the Poincaré section, $\Sigma$, as discrete dynamics. Figure 8a shows the set of points on the Poincaré section under a periodic input condition corresponding to Figure 7a. Figure 8b shows the set of points on the Poincaré section during a switching input condition corresponding to the Figure 7b, and Figure 8c shows the ellipse of the constant distance by using the mean and $\pm 1$ *S.D.* for the switching input condition. The characteristic configurations of the four clusters of sets on the Poincaré sections, that is, BF, FF, BB and FB from the top, corresponded to the Cantor set with rotation.

Figure 8d shows the return map of the Cantor set, with rotation leaving from the initial state, $x_0$, when the first input was the backhand side, the next state, $x_{1B}$, and returned to B. When the second input was the backhand side, to the next state, $x_{2B}$, returned to BB. In contrast, when the second input was the forehand side, the next state, $x_{2F}$, returned to BF. As a result, we could obtain the characteristic configurations of the four clusters at $x_2$ in Figure 8e corresponding to the empirical data. The behavior of the system was understood as the time evolution of the Cantor set with rotation (Figure 8e). In other words, the time evolution of the striking action corresponding to the two input types was characterized as a fractal transition of the Cantor set. These findings suggest that a Poincaré map analysis reveals the underlying simple rule related to exploitation of inertia in complex striking action.

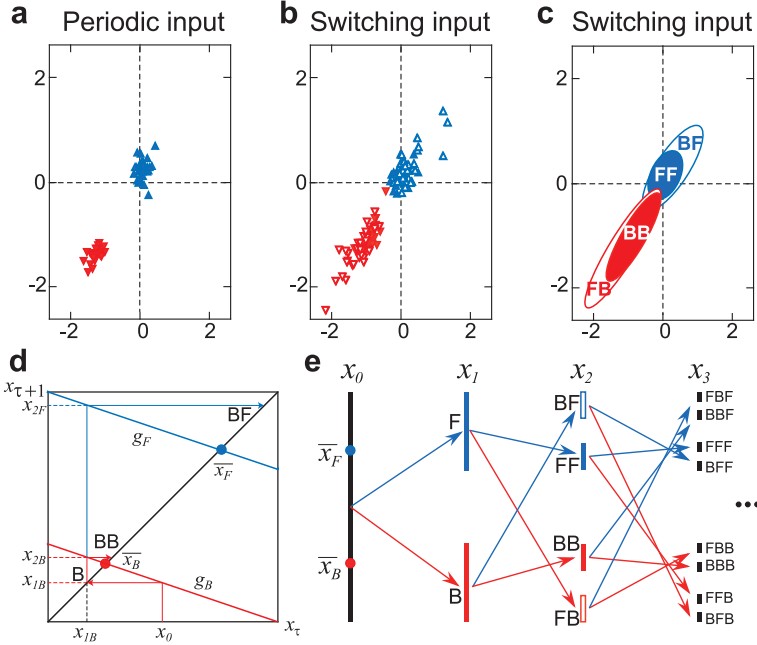

**Figure 8.** Examples of Poincaré sections $\Sigma$ for periodic and switching input conditions. (**a**) Periodic input, (**b**) switching input, and (**c**) the ellipse of constant distance using each mean and $\pm 1$ *S.D.* for switching input. (**d**) The return map shows the construction of the Cantor set with rotation using two iterative functions. The iterative functions $g_F$ and $g_B$ transform the state, $x_\tau$, to the next state, $x_{\tau+1}$. The transformations of the iterative functions $g_F$ and $g_B$ are rotated around the fixed points $x_F$ and $x_B$, respectively. (**e**) The hierarchical structure of the fractal corresponds to the sequence of forehand (F) and backhand (B) inputs. Modified with permission from [40], Hum. Mov. Sci. 2000.

## 5. Switching Hybrid Dynamics

As mentioned above, the coordination modes of interpersonal competitive behavior can be examined as a synchronization phenomenon of two-coupled nonlinear oscillators or as continuous dynamics. However, interpersonal competitive behavior shows abrupt changes in coordination modes, which differs from rhythmic interlimb coordination (e.g., [1]) or interpersonal coordination (e.g., [6]). The distribution of the relative phase region analysis reveals the characteristics of the global coordination mode without considering time evolution. Thus, it does not describe the local

coordination patterns on a shorter time scale. To solve this problem, the interpersonal competitive patterns on a short time scale were classified using a return map analysis, referring to the Lorenz map and reducing the dimensionality from continuous dynamics to discrete dynamics. Furthermore, the switching among patterns was determined by state transition probabilities and revealed the characteristics of experts. This finding suggests that continuous interpersonal competitive behavior, which seems to be a complex phenomenon, includes both perception of each other and the player's own decision making, and the two players execute their actions depending on their decisions. In other words, interpersonal competitive behavior can be regarded as a continuous switching pattern on a shorter time scale. In martial arts, such as boxing or fencing, the two players can move freely around each other. Thus, the behavior of a system comprised of two players is a weakly coupled oscillator system [8,26,33]. Thus, collective variables can be defined to describe the state of the system during continuous dynamics (Figure 3). Underlying simple rules can be identified in interpersonal competitive behavior using a return map as discrete dynamics (Figure 5). This perspective shows the interactive behavior as a whole system.

Another perspective to understand interpersonal competitive behavior is to view part of the system in the whole by focusing on individual behavior. Then, other movements are regarded as external input patterns or environmental changes, and the individual would generate their output patterns according to their input patterns. This means that the system is considered non-autonomous. The switching dynamics model suggests that the output pattern would be generated by switching among several input patterns [38,39]. Behavior during court-net sports is constrained by the movement of the ball hit by the opponent. In this case, each individual is a non-autonomous system, and the underlying simple rules could be identified as discrete dynamics (Figure 8), and complex individual behavior, as continuous dynamics (Figure 7).

The proposed integrated model is the switching hybrid dynamical system [41]. Here, we assume a system with a higher module and a lower module, which interact with each other by switching inputs from the higher to the lower module, and by using a feedback signal from the lower to the higher module. In addition, external input feeds into the higher module, as shown in Figure 9.

$$\text{Discrete dynamical system:}\quad I_l(t); l \quad = \quad \sigma(I_{ext}(t), x(t)) \tag{4}$$

$$\text{Continuous dynamical system:}\quad \dot{x} \quad = \quad f_l(x(t), I_l(t)) \tag{5}$$

Here, the discrete dynamical system $I_l(t)$ is the higher module corresponding to the brain and prefrontal cortex [42,43], and the continuous dynamical system $\dot{x}$ is the lower module corresponding to the human motor system. This system focuses on individual **A** during competition between **A** and **B**. The higher module transforms into human movement based on the continuous output pattern from the opponent **B**, $I_{ext}(t)$ and the final state of the lower module $x(t)$. The higher module considers one of three patterns $I_l(t)$ and transforms into movement. Thus, when the movement pattern of opponent **B** switches among the three patterns, the movement patterns of **A** will show three fractal trajectory subsets. However, the pattern determined by **A** is not always consistent with the continuous output pattern from opponent **B**, $I_{ext}(t)$. Because higher brain functions, such as selective attention [44–47], visual search strategy [48–50], and decision making [51–53] are redundant due to neuronal redundancy of cell assemblies [54], the same external input $I_{ext}(t)$ does not always generate the same decision $I_l(t)$. This problem might be related to expertise.

When the external inputs feed into the system, system outputs of the continuous dynamical system are generated regularly depending on continuous switching of external inputs $I_{ext}(t)$ and the final states of the system $x(t)$, as shown in Section 4.2. This suggests that the behavior of the system shows hysteresis and can be used to predict the next state. However, it has been confirmed that the regularities differ according to the length of time of the external input [55–57]. In this case, we focused on individual behavior during an interpersonal competitive situation; that is, we regarded the system as non-autonomous.

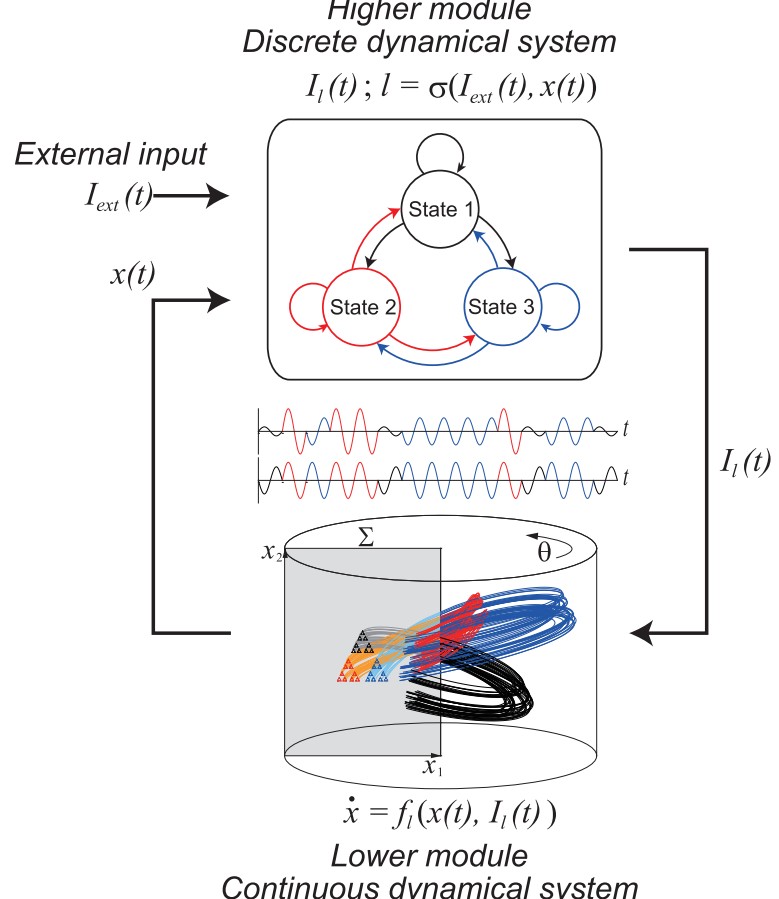

**Figure 9.** Schematic representation of switching hybrid dynamics, which is composed of a discrete dynamical system as a higher module and a continuous dynamical system as a lower module with a feedback loop. This system is non-autonomous.

When these two systems are connected to each other, it is regarded as an autonomous system as a whole (Figure 10). That is, the final state of the other system, $x_A(t)$ in **A** and $x_B(t)$ in **B** transforms into external input for the system $I_{Bext}(t)$ in **B** and $I_{Aext}(t)$ in **A**, respectively. As a result, the two systems are connected through external inputs. Then, the behavior of the whole system is described as: $\dot{X} = F(X), X = (x_A, x_B)$. In the case of kendo matches, the behavior of the whole system has been described as the instantaneous relative phase difference of the step toward-away movements of the two players. However, six offensive and defensive maneuver patterns have been found, and these patterns switch continuously during a kendo match, suggesting that the regularity underlying switching among competitive patterns could be clarified if these patterns are regarded as output patterns and/or external input patterns. To this end, we need longer time windows to observe switching among combinations of several patterns, compared to that needed for clarifying each pattern.

However, the regularity remained unclear after determining the state transition probabilities for the competitive patterns. Applying switching hybrid dynamics to interpersonal competitive behavior would help to clarify how the behavior will be generated during the time course as a whole system.

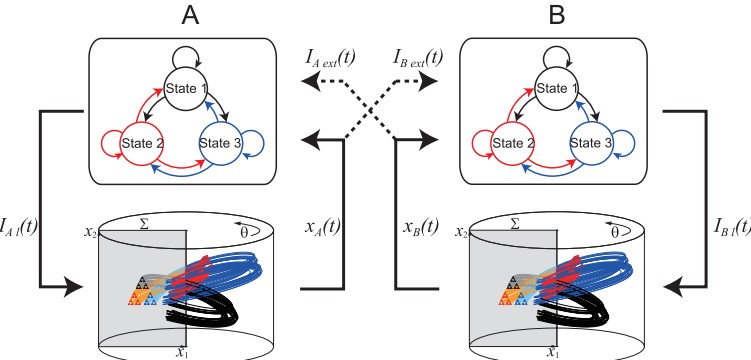

**Figure 10.** Schematic representation for two-coupled switching hybrid dynamics. This system is autonomous.

## 6. Conclusions

We investigated the manner of examination of underlying dynamics of complex interpersonal competitive behavior during sport activities using the continuous dynamical system, which was described by differential equations, and the discrete dynamical system, which was described by difference equations and/or iterated maps. Weakly coupled sports, such as boxing, fencing, and kendo, in which the players move relatively freely regardless of the opponent's movement, were examined using a two-coupled nonlinear oscillator model. Then, the order and control parameters were identified and the coordination modes between the two players were determined. Furthermore, because these continuous dynamics could be reduced to discrete dynamics using iterated maps, the coordination patterns in interpersonal competitive behavior could be depicted on shorter time scales. On the other hand, in strongly coupled sports, such as court-net sports in which movements of the players are constrained by movement of the ball hit by the opponent, the regularities in the evolution of the system were clarified using the switching dynamical system with external temporal input, which reduced the dimensionality based on the Poincaré map.

To understand the complex interpersonal behaviors such as sporting activities, we need both non-autonomous and autonomous dynamical system perspectives that propose the switching hybrid dynamical system composed of discrete and continuous dynamical systems. The proposed switching hybrid dynamical system was applied not only to court-net sports, such as tennis or table tennis, but also to weakly coupled sports, such as boxing or fencing, to understand the regularities underlying the interpersonal competitive behavior. However, further theoretical and behavioral examinations will be needed. Additionally, we plan to study the applications to the team sports, which require both intra-team coordination and inter-team competition, in future studies.

**Author Contributions:** Y.Y., A.K., M.O., K.Y. and K.G. contributed equally to all aspects of this manuscript, including idea generation, the theoretical framework, and writing.

**Funding:** This work was partially supported by Grants-in-Aid for Scientific Research (A 24240085 and 25242059) from MEXT, Japan and a Grant-in-Aid for Challenging Exploratory Research (16K12994) from JSPS, Japan.

**Acknowledgments:** We thank Kazutoshi Kudo for helpful comments and discussions that improved the manuscript.

**Conflicts of Interest:** The authors declare no conflict of interest.The founding sponsors had no role in the design of the study; data collection, analyses, and interpretation; writing of the manuscript, and the decision to publish the results.

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
