# Peer review of "A Switching Hybrid Dynamical System: Toward Understanding Complex Interpersonal Behavior"

_applsci, doi:10.3390/app9010039_

Reviewer 1 Report

This is an excellent review of Yamamoto and colleagues previous work on interpersonal coordination dynamics and switching hybrid dynamical modelling. The review not only provides a comprehensive argument for the modelling approach proposed, but provides a new detailed overview of the framework and how it can be employed to understand interpersonal and multiagent coordination in cooperative and competitive settings. 

The paper is perhaps overly terse in places, but overall the paper is well written and easy to follow. The figures are excellent.

I have no major/minors concerns. 

Author Response

I have no major/minors concerns. 

 Thank you for your kind review.

Reviewer 2 Report

The manuscript summarizes work on an interesting approach to formalizing human movement and movement in interaction with a partner or opponent where continuous dynamics are interrupted or transition from one pattern to another in a non-trivial way. The proposal is intriguing and potentially very useful as often we find ourselves limiting our experimental paradigms to extremely simplified movements simply because we are not sure how to deal with the complexity of switching patterns. Having said that, there are some issues that I believe can be ironed out with some careful re-thinking and re-writing.

The work of Jun Tani seems very relevant here and it can enrich the current manuscript as it delves with switching dynamics and how a robot can switch actions even without so-called prefrontal cortex planning just on the basis of external changes affecting its sensorimotor loop.

l. 7-8: I can't follow the connection here: "... six coordination patterns ... . ... two states ".

l. 36: tree frogs, not three frogs

l. 39: dynamics; that is, continuous dynamics -> dynamics, that is continuous dynamics

a phenomenon, not a phenomena

l. 31 and thereafter: synergetics, not synergistics

l. 42: Isn't coin flips an example of a system that's discrete in time but also with a discrete state space. Which of these are you discussing in the this paragraph?

l. 59: I wouldn't call it famous, it doesn't sound like the right word. Maybe well-known.

l. 86: This is misleading. The Hilbert transform is not applicable to an arbitrary time series. It still has to a be narrow-band signal, albeit no necessarily periodic.

Figure 1, and lines 111+: In this sort of cases the identification of the right frame of reference is crucial in determining what counts as in- and anti-phase. This reminds me of several experiments where the symmetry of coordination was manipulated to determine whether, for instance, phase stability is driven by biomechanical (homologous muscles) or perceptual coordinate frames. In the case of kendo, you count "attack" and "retract" as the relevant frame of reference. But biomechanically it's the reverse: if you attack and I retract then it means that my force vector is following your force vector in-phase. That's just a note, I am not suggesting that the one definition is to be preferred to the other.

Section 4.1 is very interesting but it comes in very abruptly. I didn't even first realize that it was a model. Then, where is the model that generated Figure 6?

l. 238. "complex human behavior". I would like to point out for the sake of generality that a wider audience could be confused by the use of human behavior. Here we're really talking of a dynamic phenomenon in the narrow sense of the word (forces applied to physical objects, back and forth) but someone could understand 'behavior' as a high-level psychological phenomenon such as decision-making. This is non-trivial since we don't know, and the manuscript doesn't give much suggestions as to what is the source of these interesting patterns. Do they emerge from the dynamics of the task or are they a manifestation of a cognitive strategy? In this sense, please be more upfront about the actual "simple rule" that you have uncovered.

Fig 8. I am not sure I followed this figure right. I don't see the Cantor set in the empirical data. Also, isn't this branching a trivial consequence of the task which requires switching, hence there's always branching per necessity, not that the branching has a particular pattern?

The model proposed in Section 5 (Eq. 4-5) leaves no causal explanatory power in the generation of movement to the environment and interaction with an opponent. This is a very classical roboticist view. Movement is a motor plant that executes internally-generated trajectories following an input-output design. How could this possibly be true given that the motor system is constantly constrained by reaction forces, stepping and maintaining balance on the ground while approaching the opponent or swinging, making contact with a ball or with the opponent, etc.? My goal is not to criticize this approach which has its background and justification. Rather, my point is that the picture presented in the manuscript is inconsistent and somewhat self-contradictory because it misses pretty much the essential theoretical aspect of the literature that it cites at the beginning, namely that movement is organized by a joint set of internal and external dynamic constraints, acting simultaneously on the moving body, not sequentially in a sense-actuate loop. Or did I miss something important?        

Author Response

Reviewer 2

The work of Jun Tani seems very relevant here and it can enrich the current manuscript as it delves with switching dynamics and how a robot can switch actions even without so-called prefrontal cortex planning just on the basis of external changes affecting its sensorimotor loop.

l. 7-8: I can't follow the connection here: "... six coordination patterns ... . ... two states ".

As you pointed out, this sentence did not clarify the relationship between six coordination patterns and two states. We added the following explanation to overcome this problem.

“In the discrete dynamical system, return map analysis was applied to the time series of movements during kendo matches. Offensive and defensive maneuvers were classified as six coordination patterns, that is, attractors and repellers. Furthermore, these attractors and repellers exhibited two discrete states. Then, state transition probabilities were calculated based on the two states, which clarified the coordination patterns and switching behavior.”

l. 36: tree frogs, not three frogs

Thank you. We apologize for the mistake and have corrected it.

l. 39: dynamics; that is, continuous dynamics -> dynamics, that is continuous dynamics

Thank you. We apologize for the oversight and have corrected it.

l. 58: a phenomenon, not a phenomena

We have made the needed corrections.

l. 31 and thereafter: synergetics, not synergistics

Thank you for pointing this out. We have rectified our mistake for both instances.

l. 42: Isn't coin flips an example of a system that's discrete in time but also with a discrete state space. Which of these are you discussing in the this paragraph?

Both. However, as you pointed out, this example might mislead the reader. Hence, we deleted this example.

l. 59: I wouldn't call it famous, it doesn't sound like the right word. Maybe well-known.

Following your suggestion, we have changed from “famous” to “well-known”.

l. 86: This is misleading. The Hilbert transform is not applicable to an arbitrary time series. It still has to a be narrow-band signal, albeit no necessarily periodic.

Thank you for raising this important point. We have used “aperiodic” instead of “arbitrary.”

Figure 1, and lines 111+: In this sort of cases the identification of the right frame of reference is crucial in determining what counts as in- and anti-phase. This reminds me of several experiments where the symmetry of coordination was manipulated to determine whether, for instance, phase stability is driven by biomechanical (homologous muscles) or perceptual coordinate frames. In the case of kendo, you count "attack" and "retract" as the relevant frame of reference. But biomechanically it's the reverse: if you attack and I retract then it means that my force vector is following your force vector in-phase. That's just a note, I am not suggesting that the one definition is to be preferred to the other.

We agree with your suggestion. The exact definition of in- and anti-phase in a coordination pattern depends on the experimental situation. We have added the definition of in- and anti-phase in the coordination pattern pertaining to kendo matches as follows:

“In this experiment, when both players stepped toward or away simultaneously, the coordination pattern was defined as in-phase coordination. In contrast, when one player stepped toward and another stepped away, it was defined as anti-phase coordination.”

Section 4.1 is very interesting but it comes in very abruptly. I didn't even first realize that it was a model. Then, where is the model that generated Figure 6?

Thank you for your positive comment about this model. This model was proposed by Gohara and Okuyama (1999a; 1999b) (cited as [38, 39]). We could not explain the details of this model because the space in the manuscript is limited. Instead, we have added the following sentence: “The details of this model have been explained in [38, 39].”

l. 238. "complex human behavior". I would like to point out for the sake of generality that a wider audience could be confused by the use of human behavior. Here we're really talking of a dynamic phenomenon in the narrow sense of the word (forces applied to physical objects, back and forth) but someone could understand 'behavior' as a high-level psychological phenomenon such as decision-making. This is non-trivial since we don't know, and the manuscript doesn't give much suggestions as to what is the source of these interesting patterns. Do they emerge from the dynamics of the task or are they a manifestation of a cognitive strategy? In this sense, please be more upfront about the actual "simple rule" that you have uncovered.

Thank you for your important suggestion. We agree this sentence may mislead a wider audience or layperson. We would like to emphasize the importance of Poincare map analysis to reveal the underlying mechanism in complex human movements, because a Poincare map analysis can reduce the number of dimensions involved in the dynamics. Also, the simple rule we found concerned the exploitation of inertia in striking action. As a result, we have changed the relevant statements as follows:

“These findings suggest that a Poincaré map analysis reveals the underlying simple rule related to exploitation of inertia in complex striking actions.”

Fig 8. I am not sure I followed this figure right. I don't see the Cantor set in the empirical data. Also, isn't this branching a trivial consequence of the task which requires switching, hence there's always branching per necessity, not that the branching has a particular pattern?

Thank you for raising this point. In the original version, we did not explain the relationship between the empirical data and the time evolution of the Cantor set with rotation shown in Figure 8. We found that the second states of the time evolution of the Cantor set with rotation in Figure 8e corresponded to the empirical data in striking action in Figure 8c (Yamamoto & Gohara, 2000). We have added the explanation about the relationship between the empirical data and the time evolution of Cantor set with rotation as follows:

“The characteristic configurations of the four clusters of sets on the Poincare sections, that is, BF, FF, BB and FB from the top, corresponded to the Cantor set with rotation.”

“As a result, we could obtain the characteristic configurations of the four clusters at x2 in Figure 8e corresponding to the empirical data.”

The model proposed in Section 5 (Eq. 4-5) leaves no causal explanatory power in the generation of movement to the environment and interaction with an opponent. This is a very classical roboticist view. Movement is a motor plant that executes internally-generated trajectories following an input-output design. How could this possibly be true given that the motor system is constantly constrained by reaction forces, stepping and maintaining balance on the ground while approaching the opponent or swinging, making contact with a ball or with the opponent, etc.? My goal is not to criticize this approach which has its background and justification. Rather, my point is that the picture presented in the manuscript is inconsistent and somewhat self-contradictory because it misses pretty much the essential theoretical aspect of the literature that it cites at the beginning, namely that movement is organized by a joint set of internal and external dynamic constraints, acting simultaneously on the moving body, not sequentially in a sense-actuate loop. Or did I miss something important?

Thank you for raising such an important question concerning our model. We point out two perspectives. The first concerns learning and performance. Human movement similar to that of a robot should solve the problem continuously imposed by environmental changes/constraints, and we could obtain the stability required to achieve goal-oriented behavior. For example, Tani (2003) showed that a mobile robot acquired internal models embedded in recurrent neural networks (RNNs); that is, he showed the existence of self-organizing attractors by learning. He distinguished the outer and internal flows. The external input in our model corresponds to the outer flow, and the feedback loop from the lower module to the higher module corresponds to internal flow. However, our model did not explain the learning or acquiring of attractors. In other words, we postulated the attractors a priori. Based on this premise, we are interested in how humans behave corresponding to abrupt environmental changes. As Tani pointed out, learning is a very interesting phenomenon. In our model, appropriate continuous movement corresponding to the external world (i.e., equation 5) and the logic that drives the movement (i.e., equation 4) would be acquired by learning. However, concerning the second perspective, we are interested in interpersonal motor skill. Then, we may consider the opponent as the environment. However, the opponent’s movement changes abruptly, corresponding to its own movement. We argue that two-coupled switching hybrid dynamics is a whole system. At the very least, we consider our model as a simultaneous, and not a sequential, model. However, human movement has inertia generated by the human’s own weight. As a result, the movement pattern is characterized by some duration, starting from the initiation of the movement. We call this movement pattern as the “state” in the higher module. We hope our explanation addresses your question.

Tani J. (2003) Symbols and dynamics in embodied cognition: revisiting a robot experiment, In Butz, M. V., Sigaud O. & Gerard P. (eds.) Mathematical foundations of discrete and functional systems with strong and weak anticipations. Springer-Verlag, 167-178.

Reviewer 3 Report

This review article provides a description of a complex interpersonal task involving kendo fencers from a dynamical systems perspective. The authors claim that the pair of individuals in a kendo match are an example of an autonomous system exhibiting hybrid dynamics that involve switching among several behavioral states (discrete dynamics) underwritten by a continuous dynamical perceptuomotor task. The review is multifaceted, complex, broad in theoretical scope, and describes several computational procedures for identifying parameters that govern the behavior of a hybrid dynamical system.

General comments:

I am not convinced that the manuscripts holds water as a standalone paper. It does read like a review article, however, but lacks the necessary details about the descriptions of dynamical systems, and glosses over many steps that would facilitate understanding by the reader. It often misfires in terms of scope: it provides extended generic descriptions of dynamical systems (e.g. Lorenz attractor) when a more focused treatment would be needed, and at other times uses too many specific terms, but without the requisite minimal explanation of what they are or how they were computed. The paper is also woefully scant on empirical details about the experiments and behavioral measurements: the participants, materials, experimental design and procedure were not provided at all. It is also not clear whether the data analysis was done on data from other previously published papers, or from the current observations. So, I was actually quite confused at times whether I should read this article as a review or as an original empirical research report. All of these shortcomings are fixable, but the authors would need to significantly revamp their manuscript.

 Specific comments:

The Introduction reads like a very generic summary of dynamical systems approach. The authors list way too many dynamical systems phenomena that are not relevant to the current project. More focused exposition is needed to motivate the present investigation. The Introduction should be utilitarian, not simply a laundry list of cool phenomena (e.g. clapping hands) without clear justification of why it is mentioned. Are all phenomena that are listed in the manuscript essential for understanding the present experiment?

Line 68: I think the correct word should be synergetics, not ‘synergistics’

Line 109-110: how was the 0.4 second requirement measured and established? A reference or short explanation might be helpful here.

Lines 111-119: Why is the switching range between 2.8 and 2.9m in kendo? Are there biomechanical reasons for this? Other?

Line 122-123: this sentence is not clear. What does learning refer to in this context?

Line 123-125: ungrammatical sentence.

Line 155: subscript in the equation for the left side term should be x+1, not x-1.

Line 177: define and explain how second-order state transition was calculated.

Figure 4: subscript in the equation for the left side term should be x+1, not x-1.

Figure 6: c) missing from text and figure.

Figure 6: the fractality of transitions is mentioned, but never explained. Why are the transitions fractal? What is the significance for theory and empirical science?

Line 302: please explain the exact time scale of what “slightly longer” time windows mean. Longer compared to what?

Line 320: the authors seem to suggest that qualitative and quantitative differences exist between court net sports that involve balls and weakly coupled sports such as boxing and fencing. Shouldn’t fencing be even more strongly coupled than ball sports since there is simultaneous physical contact between sword(dowel) and BOTH players, whereas in ball games the ball can be in the air for long time without contact by any of the players? Perhaps a table for categorizing different sports/activities/tasks with respect to the dynamical behavior they permit would be useful to include somewhere in the manuscript.

 Author Response

Reviewer 3

The Introduction reads like a very generic summary of dynamical systems approach. The authors list way too many dynamical systems phenomena that are not relevant to the current project. More focused exposition is needed to motivate the present investigation. The Introduction should be utilitarian, not simply a laundry list of cool phenomena (e.g. clapping hands) without clear justification of why it is mentioned. Are all phenomena that are listed in the manuscript essential for understanding the present experiment?

We understand your concern. However, we believe that a broader audience will not be able to understand dynamical system perspectives and human behavior without short reviews of the two-coupled oscillators model in the second paragraph, group synchrony in the third paragraph, and discrete dynamics in the fourth paragraph. These three paragraphs explain the relationship between dynamical system perspectives and previous research on human motor behavior (especially sporting activity) and not physical and/or natural phenomena. Therefore, although we appreciate your viewpoint, we would like to retain the review as is.

Line 68: I think the correct word should be synergetics, not ‘synergistics’

Thank you. We apologize for the oversight and have made the needed changes.

Line 109-110: how was the 0.4 second requirement measured and established? A reference or short explanation might be helpful here.

Thank you for your suggestion. We measured the average time of a striking action in kendo matches, and we obtained a value of 0.36 s ± 0.08 s from our previous research. We have added the reference, i.e., [34].

Lines 111-119: Why is the switching range between 2.8 and 2.9m in kendo? Are there biomechanical reasons for this? Other?

Thank you for raising this important question. Yes, there are biomechanical and tool constraints. In kendo, both players have a bamboo sword (shinai) and a point (ippon) is earned when an accurate strike is made on the opponent with the uppermost third, or top 0.3-0.4 m of the total length of the shinai (1.2 m). As a result, the average possible striking length was 2.65 m in the experiment [34]. Accordingly, the switching range of between 2.8 and 2.9 m was slightly longer than this average possible striking distance, in line with the not-to-lose strategy. We have added a short explanation concerning this reason as follows:

“This switching range (2.8-2.9 m) was slightly longer than average possible striking distance, in line with the not-to-lose strategy [34].”

Line 122-123: this sentence is not clear. What does learning refer to in this context?

We agree your comment. We have not explained what was learned in this experiment. We have added a short explanation as follows:

“In addition, we examined the learning process in interpersonal competition via the play-tag game [35]. The relative phase analysis revealed that the synchronization modes changed from in-phase to anti-phase coordination by learning, because the players learned the not-to-lose strategy.”

Line 123-125: ungrammatical sentence.

Thank you for your comment. We have changed this sentence as follow;

“However, in this analysis, the distribution of the relative phases across nine 20° relative phase regions, from 0° to 180°, was calculated. It means that the tendencies in pooled data can perceive, but cannot reveal, the time evolution in the time series data.”

We hope you find these changes acceptable.

Line 155: subscript in the equation for the left side term should be x+1, not x-1.

We have written it as .

Line 177: define and explain how second-order state transition was calculated.

We have provided the definition of the second-order transitions. For clarification, we have also changed the order of sentences as follows:

“We identified four trajectories, namely, { = F, = F}, { = N,  = N}, { = F,  = N}, and { = N,  = F}, as second-order transitions. Figure 5 shows the second-order state transition diagrams for experts and intermediates.”

 Figure 4: subscript in the equation for the left side term should be x+1, not x-1.

We have written it as .

Figure 6: c) missing from text and figure.

Thank you. We apologize for the error. We have corrected it. We have also added a reference to Figure 6c in the text.

 Figure 6: the fractality of transitions is mentioned, but never explained. Why are the transitions fractal? What is the significance for theory and empirical science?

We agree we have not explained the fractality of transitions. In this example, three inputs are switched stochastically with finite intervals, and the trajectories corresponding to the input cannot converge to the excited attractors because there is insufficient time to converge. As a result, the trajectories spread out around the excited attractors. The cross points on the Poincare section also spread out around the cross points, corresponding to the excited attractors, with a fractal-like structure. The details of the computational experiments are shown in Gohara and Okuyama (1999a; 1999b), cited as [38, 39]. The significance for the theory and empirical data concerns switching among several excited attractors within finite intervals, because human behavior, especially interpersonal coordination or competition in sporting activity, can switch abruptly, and not maintain cyclic or rhythmic movement. However, we can coordinate with others. Thus, we need to consider the switching dynamics as a non-autonomous system. We agree that the explanation about fractal transition is insufficient. However, we have limited space in this manuscript. We have added a brief explanation about fractal transitions as follows:

When the three inputs are switched stochastically into the system with finite intervals, the trajectories switch corresponding to the input (Figure 6b). It is known that these trajectories spread out around the excited attractors with a fractal-like structure. As a result, the Poincare section shows the Sierpinski gasket…The details of this model have been explained in [38, 39].”

 Line 302: please explain the exact time scale of what “slightly longer” time windows mean. Longer compared to what?

We agree your suggestion. Here, we would like to mention switching among several patterns observed in kendo matches, and these patterns correspond to the external input pattern in two-coupled switching hybrid dynamics. Because each pattern has a particular duration, we need a “slightly longer” time window to observe switching among combinations of several patterns, compared to that needed to observe each pattern. We have added this explanation as follows:

However, six offensive and defensive maneuver patterns have been found, and these patterns switch continuously during a kendo match, suggesting that the regularity underlying switching among competitive patterns could be clarified if these patterns are regarded as output patterns and/or external input patterns. To this end, we need longer time windows to observe switching among combinations of several patterns, compared to that needed for clarifying each pattern.”

Line 320: the authors seem to suggest that qualitative and quantitative differences exist between court net sports that involve balls and weakly coupled sports such as boxing and fencing. Shouldn’t fencing be even more strongly coupled than ball sports since there is simultaneous physical contact between sword(dowel) and BOTH players, whereas in ball games the ball can be in the air for long time without contact by any of the players? Perhaps a table for categorizing different sports/activities/tasks with respect to the dynamical behavior they permit would be useful to include somewhere in the manuscript.

Thank you for your valuable comment. We agree with your comment partially. We could regard fencing as a strongly coupled sport and court-net sports as weakly coupled sports. For an exact categorization, we need to categorize different sports activities from the dynamical system perspective, as you pointed out. We have defined weakly coupled sports and strongly coupled sports in the previous paragraph. In court-net sports, one player has to respond or chase a ball hit by an opponent. This means that a ball hit by an opponent corresponds to the external input in switching dynamics. On the contrary, fencing or boxing was regarded as a weakly coupled sport in the two-coupled oscillators model. Hence, we have proposed that the switching hybrid dynamical system could apply to weakly coupled sports in the last paragraph.

Round  2

Reviewer 3 Report

For the most part the reviewers addressed my comments. However, I am still not completely sold on the merits of the article. I would like to see a paragraph that details what this review article accomplishes besides rehashing past findings from empirical studies. Specifically, I would like to see a more forceful summary that details how knowledge gained from several reviewed empirical studies has been integrated and what each study contributed to the understanding of the dynamics of interpersonal sports. Is it the distinction between autonomous vs. non-autonomous dynamical characterization? Is it the categorization of weakly-coupled vs strongly coupled activities? Something else?

The manuscript would still benefit from a thorough review by an English grammar expert.

Line 93: the word “as” is missing after “used”.

Line 133-34: the word “perceive” does not make sense in the sentence.

Figure 8 d) ungrammatical phrase “Manner in which shows”

Line 344: “iwriting” has a typo.

Author Response

For the most part the reviewers addressed my comments. However, I am still not completely sold on the merits of the article. I would like to see a paragraph that details what this review article accomplishes besides rehashing past findings from empirical studies. Specifically, I would like to see a more forceful summary that details how knowledge gained from several reviewed empirical studies has been integrated and what each study contributed to the understanding of the dynamics of interpersonal sports. Is it the distinction between autonomous vs. non-autonomous dynamical characterization? Is it the categorization of weakly-coupled vs strongly coupled activities? Something else?

 Thank you for your constructive comments. We agree with your comment on the lack of a strong message from this review article. Therefore, in the revised manuscript, we have added a summary, in the Conclusion section, obtained from our several empirical studies, as follows.

“To understand the complex interpersonal behaviors such as sporting activities, we need both non-autonomous and autonomous dynamical system perspectives that propose the switching hybrid dynamical system composed of discrete and continuous dynamical systems.”

 The manuscript would still benefit from a thorough review by an English grammar expert.

 Line 93: the word “as” is missing after “used”.

 Thank you for your comment. Following your suggestion, we have added “as” after “used,” as follows.

 “In addition, the speed scalar has been used as a collective variable to describe different patterns during badminton [31]”

 Line 133-34: the word “perceive” does not make sense in the sentence.

Thank you for your comment. We have changed this sentence to clarify our message, as follows.

 “It means that the tendencies in the pooled data can be revealed, but the time evolution in the time series data cannot.”

 Figure 8 d) ungrammatical phrase “Manner in which shows”

 Thank you for your comment. As you pointed out, this sentence was grammatically incorrect, and has been changed as follows.

 “The return map shows the construction of the Cantor set with rotation using two iterative functions.”

 Line 344: “iwriting” has a typo.

 Thank you for your comment. We apologize for this oversight and have now corrected it.
